# The Effect of Company Ownership on the Environmental Practices in the Supply Chain: An Empirical Approach

Liliana Rivera [1], Norma Ortiz [1,*], Gabriel Moreno [1] and Iliana Páez-Gabriunas [2]

1   School of Management, Universidad de los Andes, Bogotá 111711, Colombia;
    ml.rivera@uniandes.edu.co (L.R.); gm.moreno10@uniandes.edu.co (G.M.)
2   School of Administration, Universidad del Rosario, Bogotá 110111, Colombia; iliana.paez@urosario.edu.co
*   Correspondence: nc.ortiz@uniandes.edu.co; Tel.: +57-3125191929

**Abstract:** Investors are increasingly drawn to ESG-based investing because they seek well-run businesses, believing that companies integrating ESG factors are better managed. However, the impact of company ownership on environmental performance remains unclear. This study aims to address this gap by examining the relationship between company ownership and company interest in measuring the environmental impact of its supply chains, as the environmental aspect is directly linked to supply chain activities and has quantifiable measures. Using random effects ordered logistic regression on panel data from 2017 to 2022 for 2811 companies, we show that companies with long-term investment sources demonstrate a greater interest in measuring environmental variables in their supply chain compared to those financed with short-term investments. Sovereign wealth funds and other long-term investment sources exhibit a positive and significant correlation with higher utilization of ESG indicators in the supply chain. These findings suggest that policymakers and private companies aiming to enhance sustainability should prioritize longer-term investment sources, which display stronger commitments to sustainability and ESG practices and are more likely to use environmental initiatives in their supply chains.

**Keywords:** sustainability; environmental supply chain management; green supply chain; ESG; ownership; financial performance; principles for responsible investment; corporate social responsibility

## 1. Introduction

During the last decades, the emergence of initiatives such as the Kyoto Protocol (1997) or the Paris Agreement (2015) has given rise to the growing discussion about the impact of human activity on environmental changes and, above all, its influence on phenomena such as climate change. Now, why should we be concerned about climate change? Mainly due to effects such as global warming, ecological imbalance, and increased natural disasters, which in turn cause economic and social problems [1]. Global headlines are focusing on how climate change is causing increasingly extreme weather patterns. In March 2023, the Intergovernmental Panel on Global Change (IPCC) released a final warning on the consequences of what it considers to be a global disaster. The warning was part of the IPPC's comprehensive assessment report, which emphasized that "human activities have caused approximately 1 °C of global warming above pre-industrial levels, with a range between 0.8 °C to 1.2 °C. Moreover, global warming is likely to reach 1.5 °C between 2030 and 2052 if it continues to increase at the current rate" [1]. This increase in temperature can bring different consequences and affect people's living standards. Among the risks of ignoring this threat are (i) an increase in temperature, (ii) a shortage and reduction in water quality, (iii) a reduction in air quality, (iv) an increased likelihood of drought, and (v) increases in sea level, among others [1].

That is why, in recent years, environmental sustainability has become an increasingly prominent issue for consumers, businesses, and investors alike. As a result, companies are

being pushed harder than ever to implement environmentally conscious procedures across their whole operation, including their supply chains. However, implementing sustainable practices frequently requires significant investment, and companies may find it difficult to strike a balance between the requirement for immediate profitability and sustainability's long-term advantages [2]. In this context, environmental, social, and governance (ESG) criteria have emerged as a comprehensive framework for assessing the performance and responsibility of companies in these areas. ESG refers to a set of criteria used to assess a company's impact on the environment, its commitment to responsible social practices, and the quality of its corporate governance. These criteria not only focus on the financial aspect of the business but encompass a broader range of factors that can influence a company's long-term sustainability and success [3].

Furthermore, the importance of sustainability investments is not diminishing. One characteristic that could sum up the state of supply chains today is the need to navigate extreme change, where supply chain sustainability is the moving target. For instance, in recent years, there has been increasing pressure from stakeholders for companies to pursue sustainable supply chain objectives (SSC) [4]. As a response, companies have been investing in environmental and social practices in supply chain management over the past 20 years [5], which is known as sustainable supply chain management (SSCM). There is evidence indicating that this pressure is also translating into action, as statistically significant relationships have been observed between how firms rate pressure from their investors and their declaration of net zero goals, where dimensions such as climate change mitigation, supply chain circularity, adoption of technology, and practices to support SCS goals show the most positive trend [4]. In particular, the number of companies that adopted climate change mitigation technologies increased in the last few years [6]. It is also important to highlight that a major part of ESG growth has been driven by the environmental component of ESG and responses to climate change. But other components of ESG, in particular, the social dimension, have also been gaining prominence.

Similarly, firms have been devoting greater resources to enhancing ESG, according to the Governance & Accountability Institute [7], across industries, regions, and businesses. In fact, almost 95% of S&P 500 firms and about 80% of Russell 1000 companies submitted ESG reports in 2022. Also worth mentioning is the significance of concentrating on supply chains, which account for more than 80% of greenhouse gas emissions and more than 90% of the effects on air, land, water, biodiversity, and geological resources [8]. These costs are significantly higher than those caused by a company's own operations. As a result, any program that wants to meet sustainability or emission duties should primarily focus on its larger network and include stakeholders in procurement and supply chain management. Moreover, businesses are increasingly including ESG data in earnings reports, and investors are increasingly drawn to businesses that make ESG investments and utilize ESG disclosures to highlight their sustainability initiatives. In that sense, as a consequence of public concerns about the pandemic, climate change, and the misuse of natural resources, investors are shifting their focus toward sustainable businesses and weeding out those with outmoded practices, such as unfair wages, investments in fossil fuels, unsustainable agricultural practices, and the production of non-recyclable products [9]. Businesses involved in ESG activities can impact investment decisions by providing investors with a thorough understanding of their practices, allowing investors to choose a firm that provides a sustainable future with a low-risk profile. In 2022, a Dow Jones poll of two hundred investment experts predicted that ESG investments will more than quadruple over the following three years, while different research indicated that 48% of investors are interested in sustainable investing funds [9].

As a result, a significant portion of the literature has concentrated on analyzing the effects of ESG on financial performance. In 58 percent of the company studies that focused on operational indicators like ROE, ROA, or stock price, Whelan et al. [10] found a positive association between ESG and financial success. They discovered that almost 60% of investment studies that focused on risk-adjusted features (alpha or the Sharpe ratio on

a portfolio of equities) demonstrated comparable or superior performance to traditional investment programs, while only 14% demonstrated adverse consequences [10]. Some ESG strategies appeared to generate market value or excess returns for investors looking to build portfolios, especially for long-term investors, and provided downside protection during economic or social downturns, compared with conventional investment strategies [10]. Additionally, arguments in favor of adopting ESG initiatives are growing, emphasizing medium and long-term benefits, including better financial performance for the companies that adopt them. This has been evidenced in studies showing that incorporating this type of strategy increases firm market value [11,12], sales [13], reputation [14], image [15], and performance [16]. Notably, very few studies found definitive negative correlations between ESG and financial performance.

In that sense, the current literature lacks research focusing on whether institutional investors with distinct types of equity funding or ownership play divergent roles in supply chain sustainability. The primary source of equity funding for a firm could be a significant variable that may have an impact on its capacity to implement sustainable practices. Different equity investors, such as pension funds, hedge funds, private equity, and sovereign wealth funds, may have different priorities and investment horizons, which could affect their willingness to support sustainable practices. Understanding the impact of different equity funding sources on a company's supply chain environmental performance is therefore a critical area of research.

For instance, there is evidence from many countries that institutional investors advocate for better ESG performance at the corporate level. In other words, businesses are improving their social and environmental performance because investors demand it [17]. They create firm-level metrics for $CO_2$ emissions, renewable energy use, human rights abuse, and employment quality and analyze whether there is a link between share ownership and ESG performance. Additionally, company ownership structure may have a direct impact on environmental performance with the creation of incentives for cost-cutting and increasing revenue, which may have a secondary effect on environmental performance with an increase in firm profitability that enables higher investments in pollution-reduction upgrades or more efficient production processes [18]. On the other hand, according to instrumental stakeholder theory, firms use corporate social responsibility (CSR) to manage their relationships with powerful stakeholders in order to obtain support or resources from them. Consequently, a firm's CSR policies are affected by the existence of greater interest from key stakeholders in sustainable development, leading to higher pressures on companies to adopt ethical and sustainable practices. Institutional investors, due to their significant role in capital markets, are considered a key stakeholder group that can exert a significant impact on CSR strategies [19]. However, institutional investors' support for the implementation of SDGs by the companies in which they invest is mainly driven by corporate complexity and, specifically, by the internationalization of business investment and by companies operating in industries with greater social impact [19].

Long-term, pressure-resistant institutional investors, such as foundations, mutual funds, investment funds, sovereign wealth funds, pension funds, and endowment funds, are more responsive to external pressures addressing environmental sustainability concerning time horizons. Because their investments have a long time horizon and seek a smaller immediate economic return, according to García-Sánchez [2], these investors will demonstrate a higher commitment to innovation and sustainability, encouraging the development of more proactive environmental practices. In a comparable way, Aibar-Guzmán et al. [20] state that institutional investors' ownership of shares with a long time horizon and strategic goals boosts the influence of these projects on a company's reputation, market value, and profitability. As a result, to the extent that these patient investors have a long-term perspective, they will favor investments that can increase a firm's long-term worth and competitive position, even when their outcomes and rewards take a long time to mature. Additionally, when investors and the companies in which they invest are independent of one another, managers can be under additional pressure to adopt creative sustainability-focused initiatives that raise the value of the businesses [2].

On the other hand, institutional investors that are susceptible to short-term pressure, such as financial institutions like banks and insurance firms, in addition to cross holdings, are less sensitive to outside forces and have a shorter-term, more economic perspective. They support a short-term focus on managers because they are investors looking for quick returns by discouraging the implementation of proactive environmental policies [2].

With that in mind, ESG has become increasingly important in investment decision-making, and investors are increasingly demanding that companies prioritize these factors in their operations. In response, companies are establishing ESG policies and practices to draw in investment and keep stakeholders' trust. However, it is unclear how different equity investor types may affect a company's decision to implement ESG principles across its supply chain. By examining the effects of equity capital origin on supply chain environmental performance and ESG adoption, this paper aimed to address that uncertainty. Specifically, this paper sought to examine how several types of equity investors, with different investment horizons, influence a company's adoption of sustainable practices in its supply chain using ESG performance evaluations. This research focused on the environmental aspect of ESG because it is most directly related to supply chain activities (the impact of resources, processes, and distribution on climate change) and because there are measures to analyze. Pension funds, hedge funds, private equity firms, and sovereign wealth funds were the main focus as these are the most common types of equity investors. To do so, a categorical logit regression with data from the Refinitiv database was conducted. The analysis focused on studying the relationship between a company's source of ownership and its supply chain's environmental performance in ESG.

Our study analyzed data on 2811 companies from various industries and locations around the world with different sizes, which were available in Refinitiv during the years from 2017 to 2022. The results indicate that companies with long-term investment sources have a higher interest in measuring environmental variables in their supply chain compared to those financed with short-term investment sources. Thus, investments with a longer-term horizon, such as sovereign wealth funds, showed a positive and significant correlation with greater use of ESG indicators in the supply chain.

This paper proceeds as follows. After this introduction, the Section 2 provide a more detailed literature review. Section 3 presents and describes data samples, and Section 4 sets out the empirical framework and model construction. Then, in Section 5 we present and discuss the main findings of this study. In the Section 6, the main conclusions of this study are drawn, and the implications of the findings are discussed.

## 2. Literature Review

### 2.1. What Is ESG and How Did It Emerge?

ESG, which stands for environment, social, and governance practices, encompasses three critical areas of focus. Recent global events, including the Russia–Ukraine conflict, the COVID-19 pandemic, the climate emergency, and the surge in economic inequality have all acted as significant catalysts for organizations to re-evaluate the kinds of impacts they want to have on different stakeholders such as employees, clients, suppliers, and the communities in which they operate [21]. Therefore, while financial performance remains a key metric of success in the private sector, businesses utilizing ESG aim to demonstrate that their goals go beyond maximizing profits. In this way, ESG provides a framework that enables organizations to measure, quantify, and report their commitments and actions toward creating a better world.

The idea of ESG indicators, according to Baid and Jayaraman [21], was included in the UNGC's "Who Cares Wins—Connecting Financial Markets to a Changing World" report from 2004 [22], and it was endorsed by 20 financial institutions, including big banks like BNP Paribas, HSBC, and Morgan Stanley; asset managers, like Henderson Global Investors; asset owners, like Allianz SE and Aviva PLC; and other stakeholders. Likewise, the "Freshfield Report", published in 2005 by the United Nations Environmental Program's Finance Initiatives (UNEP-FI), provided the first indication of the financial

significance of ESG issues and covered in full the fiduciary duty associated with the use of ESG data in investment decisions. For instance, the UN-sponsored Principles for Responsible Investment (PRI) project, which was launched in 2006 and has attracted financial institutions as signatories, was founded on these two publications. Finally, the rise in PRI signatories was a definite indication of the importance, demand, and inclusion of ESG mandates among investors as well as their growing knowledge and demand [23,24].

### 2.2. Sustainable Supply Chain Management

Given the rising concerns about environmental and societal issues, businesses are increasingly integrating sustainability practices into their processes and strategy for supply chain management in order to be aligned with sustainability goals [25]. SSCM is defined as a management process that integrates environmental considerations, social performance, and economic contributions [26], and it seems that doing so has paid off for companies [27], as SSCM generates, in general, positive results both for the organizations that carry it out [5] and for society and the environment [13].

According to Khodakarami et al. [28], when businesses integrate sustainability goals into their core functions, they achieve a better market position in the global context. It has been demonstrated that companies that use sustainable practices in their supply chain management tend to be leaders in the market and experience long-term economic benefits [29]. Pagell and Wu [30] argue that to be successful in implementing sustainable strategies, companies must consider simultaneously the integration of the best traditional organizational practices in the supply chain along with new sustainable practices.

It is imperative to recognize that ESG issues can arise across various supply chains, extending beyond their current relevance for businesses. Baid and Jayaraman [21] emphasize the importance of establishing a robust framework for identifying and managing supply chain problems, as well as implementing active governance, risk management, and corrective processes. The supply chain plays a central role in ESG considerations. Corporate social responsibility (CSR) and supplier security are closely intertwined, and this relationship has implications for the entire supply chain, as noted by Tiwari et al. [31]. In order to foster socially responsible behavior throughout the entire supply chain, Aras and Crowther [32] found that socially responsible organizations hold themselves accountable for the performance, productivity, and welfare of all their suppliers.

### 2.3. SSCM Challenges in the Current and Global Context

There are presumptions in the literature regarding the importance of ESG for the generated value from the supply chain; however, certain research has revealed that the vast majority of investors are more interested in measuring and reporting ESG data for financial reasons than just ethical ones. Although the extent of the data varies constantly between countries, industries, and even business models, one study found that the majority of respondents in all polls said that ESG data were crucial to the success of investments [33]. Large, complex, international, and for-profit firms that are incorporating the difficulty of managing social, environmental, and financial performance into decision-making are undergoing a paradigm change [34].

In order for the supply chain to achieve improved ESG performance, some hurdles must be overcome, as previously identified in the literature, mostly in relation to the geographical dispersion of firms. According to Baid and Jayaraman [21], supply chains frequently have numerous tiers and cross borders, and outsourcing and offshore further increase their complexity and transparency. The fact that supply chains are not a company's core business exposes them to significant risks from ESG issues like child labor, human rights abuses, corruption, environmental challenges, and many others. The design of a supply chain frequently takes into account service, quality, and cost/time. This has served as the foundation of supply chains for many years and continues to be effective for both customers and supply chain specialists. Nonetheless, this has resulted in activities that are not sustainable, such as high carbon emissions, the effects of climate change, unfair

practices, labor problems, wealth disparity, and, to top it all off, bad governance that leads to questionable ethical behavior in firms. Throughout time, organizations have addressed environmental problems, but social issues are still in their infancy with very little progress.

Finally, a rising understanding of the relationship between products and services and quality of life is being facilitated by the regulatory environment, policies, standards, and regulations for ESG, particularly in Europe, the United Kingdom, and Australia. In that sense, organizations must make every effort to integrate ESG activities into their supply chains because of the impending changes [21].

### 2.4. Importance of the E over the S and the G in ESG

Sustainable strategies have been shown to be cost-effective in the product life cycle [35,36]. In particular, the adoption of practices oriented at taking care of the environment—also called green practices—has been shown to generate cost benefits for companies related to reduced energy consumption [37], packing costs [29], lower labor costs [26,38], and waste [39], among others.

Furthermore, according to Hall and Matos [40], the social aspect of sustainability is one of the most important aspects of SSCM, as it involves relationships with multiple stakeholders. However, the focus specifically on socially responsible supply chain management behavior and practices and their influence on organizational performance has been very limited [41].

Concerning the governance aspect of ESG strategies, sustainable supply chain governance is understood as a set of practices and initiatives oriented to strengthen the relationships between an organization and its supply chain actors and stakeholders [42].

### 2.5. The Importance of Sustainability

Sustainability has gained increasing relevance among companies and investors, primarily due to the incentivization provided by the UN Paris Agreement, which emphasizes the consideration of environmental, social, and governance (ESG) initiatives. This global agreement has led to the promotion of sustainable finance, prompting companies and investors to integrate ESG goals into their investments, which involves the integration and balance of economic, social, and environmental outcomes [43]. On the other hand, responsible asset management involves making ownership and investment decisions that take into account ESG factors. To showcase their dedication to responsible investing, many investment managers endorse the United Nations Principles for Responsible Investment (PRI). This endorsement serves as evidence of the increasing significance placed on responsible investment, as indicated by the substantial assets under the management of PRI signatories, which reached USD 121.3 trillion in 2021 [44].

The integration of ESG considerations into firm operations contributes to sustainability [45], which, in turn, enhances a company's appeal to investors. Institutional investors have increasingly prioritized environmental issues, particularly climate change, when making investment decisions [20]. Consequently, there has been a notable increase in the adoption of climate change mitigation technologies by companies in recent years, driven by strategic objectives and the interest of institutional investors with long-term investment horizons [6]. In response to institutional pressure, investor-owned firms have undertaken a greater number of ESG initiatives [46]. This trend is reflected in the fact that in 2020, 24% of companies incorporated ESG into their corporate strategy [6]. Moreover, traditional investors have also recognized the significance of ESG factors in investment decisions, with 72% of them incorporating these factors into their investment processes in 2022 [47].

Among the three dimensions of ESG (environmental, social, and governance), the environmental dimension is widely recognized as crucial for sustainable growth [48]. Moreover, the case for adopting ESG initiatives continues to strengthen, highlighting the medium and long-term benefits, including improved financial performance, for companies that embrace sustainability strategies. Achieving a balance between financial objectives and sustainable development with ESG initiatives is increasingly recognized as necessary and

advantageous for companies [49]. By maintaining consistency between financial and ESG growth, companies can create a balanced portfolio of initiatives that generate sustainable growth and long-term value.

The relationship between an organization's type of ownership and its performance regarding ESG indicators is a topic that remains underexplored in the literature. Understanding why increasing amounts of capital are being invested in companies with better ESG criteria, depending on the type of investor, is crucial. It is also important to assess the associated risks and examine whether major institutional investors, such as pension funds, hedge funds, sovereign wealth funds, and insurers, genuinely prioritize transparent and careful management of their client's investments in companies with higher ESG ratings. Therefore, it is essential to review existing analyses in the literature regarding the interactions between key investor groups and ESG management, including the issues and challenges explored in each context.

By examining the existing literature on sustainability and ESG integration into business, we can gain insights into the growing importance of ESG factors for companies and investors, the positive effects of sustainability strategies on financial performance and reputation, and the need for a balanced approach that aligns both financial objectives and sustainable development. In this way, exploring the relationship between different investor types and their engagement with ESG initiatives will provide valuable insights into the motivations and challenges associated with integrating sustainability considerations into investment decisions.

### 2.6. The Case of Insurers and Pension Funds

Concerning investors, they can influence a company's decision to adopt practices aimed at meeting ESG goals. Long-term investors, such as pension funds, are known to incorporate companies with clear ESG objectives and track records into their investment portfolios. Research indicates that institutional investors with long-term investment horizons, like pension funds [46], tend to prioritize environmental and social investments more than short-term institutional investors, such as mutual fund investors, who have varying investment objectives over different timeframes [2].

The presence of long-term institutional investors with strategic objectives, as highlighted by Aibar-Guzmán et al. [20], can significantly enhance the impact of ESG projects on a company's image, market value, and profitability.

Notably, insurance firms and pension funds are among the largest institutional investors globally [50]. These institutions play a vital role in addressing sustainability challenges with investment management practices. When evaluating their organizations, insurance firms and pension funds consider ESG issues to assess financial risks and potential. Aburto Barrera and Wagner [51] found that although developing complex scenarios and models may pose challenges, the insurance industry is adept at envisioning how natural disasters could impact risk management and underwriting in the future. The significance of insurance companies and pension funds in resolving ESG issues is acknowledged by the UN Environment Program (UNEP, 2012) [52]. Their involvement is considered essential in tackling ESG challenges effectively. Their investment decisions and consideration of ESG factors can drive positive change and contribute to the broader sustainability agenda.

According to Boffo and Patalano [53], investors have predominantly relied on ratings to transform ESG data into investment products and capitalize on this knowledge. However, as ESG practices gain traction in the insurance sector, the need for the development of ratings persists. Presently, market data providers such as Thomson Reuters, Bloomberg, Morningstar, MSCI, and Sustainalytics are addressing this demand by targeting financial institutions. These providers take into account crucial aspects across the three dimensions of ESG. Within the social component, concerns related to diversity, health, human rights, privacy, and community involvement are considered. Environmental factors encompass considerations such as carbon emissions, pollution, and the sustainable use of natural

resources. And, finally, the corporate aspect covers areas such as shareholder rights, board independence, and corporate ethics.

Furthermore, although major pension funds actively strive to enhance their ESG procedures, there is always room for improvement. The incorporation of ESG considerations into investment management is crucial for achieving long-term investment returns and addressing climate-related risks [54]. Notably, pension funds are increasingly advocating for investments in companies that demonstrate social and environmental objectives [55]. This trend is particularly evident in the European Union (EU), where investment policies prioritize the impact of environmental, social, and governance factors on pension funds [55]. In contrast, other regions with common law jurisdictions, grounded in trust law, tend to prioritize investment returns over ESG considerations. As it is, the EU's regulatory framework for pension funds emphasizes the promotion of ESG goals, with decisions regarding these goals guided by financial considerations.

### 2.7. The Case of Hedge Funds

Hedge funds are increasingly incorporating ESG investment strategies, although their approaches vary significantly. According to the OECD's 2020 survey, 46% of respondents currently integrate responsible investing criteria into their equity long/short strategies, and the majority (65%) plan to do so within the next two years, as indicated by studies conducted by Cerulli and UNPRI. The use of ESG investments is growing alongside the evident desire for leverage and alpha investing opportunities [53]. In subsequent investigations, Liang et al. [44] revealed that 65% of hedge fund investors surveyed in a 2018 Preqin poll believed that ESG considerations would become increasingly significant. However, only 37% of hedge fund managers shared this viewpoint. This disparity in perspectives suggests that hedge fund managers may embrace responsible investing practices to align with investor preferences and demands.

Given their lack of transparency, disclosure, and regulatory monitoring, according to Liang et al. [44], hedge funds are especially prone to agency issues and predisposed to opportunistic conduct. However, they remain a crucial component of responsible institutional investor portfolios. According to agency theory, hedge funds with low-ESG signatories exhibit various manifestations of management opportunism. These funds were more likely to display abnormal patterns in reported fund returns, which could indicate fraudulent activities, and they had a higher propensity to engage in regulatory violations, risky investments, and severe infractions. Conversely, the results were better for low-ESG signatory funds than for high-ESG signatory funds, as an agency theory would suggest [44].

And lastly, Liang et al. [44] found that hedge funds supported responsible investing profit in real and financial ways. After adjusting for the usual culprits, funds that supported responsible investment bought in a lot more money than other funds. In general, investors do not differentiate between signatories with poor and high ESG ratings. Low-ESG signatories draw investment by aggressively marketing their funds to novice investors like HNWIs, who are less able to appropriately assess ESG exposure.

### 2.8. The Case of Sovereign Wealth Funds

Sovereign wealth funds (SWFs) are state investment funds managed by governments for macroeconomic purposes, aiming to secure and increase national wealth in the long term. These funds handle both national and international financial assets, primarily derived from gas and oil exploitation, bonds, shares, and real estate, making them long-term investment vehicles.

The investment strategies for SWFs vary and depend heavily on government policies. However, they often face criticism for lacking transparency in their processes, as evidenced in the Sovereign Wealth Fund Institute (SWFI) investment reports from 2008, leading to suspicion regarding their operations. Liang and Renneboog [56] highlight that most SWFs are not perceived as transparent, as they provide minimal information about their environmental, social, and governance (ESG) operations and policies. Despite not actively

promoting ESG initiatives, SWFs tend to integrate ESG factors into their investments [57] and consider stakeholder perspectives in their investment goals [56]. Including stakeholder interests, such as corporate social responsibility (CSR) and ESG initiatives, and pursuing socially responsible investments are practices recognized for their positive impact on long-term investment performance, reputation, and risk mitigation [58–60]. Consequently, firms with high ESG and CSR indicators tend to attract investments from SWFs seeking secure and sustainable financial returns in the long run [61,62].

*2.9. Individual Investors*

The Global Sustainable Investment Alliance [63] report highlights the top ESG concerns for individual investors, with product quality and financial fraud identified as the most significant areas. Additionally, safety concerns and environmental violations emerge as factors influencing investment decisions, as reported by the majority of respondents. In that sense, the report encompasses surveys conducted by various organizations within different countries, providing valuable insights into investor perspectives on sustainable investing.

A study conducted by Morgan Stanley in 2019 [64] revealed that in the United States, 85% of individual investors surveyed expressed interest in sustainable investing, marking a 10% increase from 2017. Notably, a staggering 95% of millennials showed interest, reflecting a 9% increase from 2017. The study identified specific areas that garnered the most attention from respondents, with plastic reduction (46%) and climate change (46%) being the top concerns. Additionally, community development (42%), circular economy (39%), and the Sustainable Development Goals (SDGs) (36%) were also significant areas of interest, in that order. In another survey conducted by Natixis Global Asset Management in 2019 [65], participants in retirement contribution plans were asked about their views on sustainable investing. A significant 75% of the respondents agreed that it was important to improve the world while simultaneously increasing their wealth. Furthermore, 61% of the participants indicated that the availability of sustainable investing options would make them more likely to contribute to retirement plans.

In contrast, ESG mutual funds have seen a substantial increase in ESG products since 2015, according to the China Sustainable Investment Forum (China SIF). In 2020, more than 20 ESG mutual funds were released. Additionally, ESG mutual fund size increased dramatically, reaching more than double the amount in 2019, representing the quickest growth ever. Moreover, according to the Survey of Public Attitudes Towards Sustainable Investment conducted by China SIF and Sina Finance in October 2020, the majority of individual investors take these factors into account when making investment decisions, even though they may not be familiar with the terms SRI or ESG. The survey highlights that a significant 86% of the respondents consider sustainable investment criteria when making investment decisions. The two primary ESG areas of concern for individual investors, as outlined in the report, are financial fraud and product quality. Moreover, more than half of the respondents indicated that they believe environmental violations and safety concerns influence their investment choices.

On the other hand, in Canada, there has been a notable rise in the utilization of designated sustainable and ethical investing products, including mutual funds and exchange-traded funds (ETFs). According to Morningstar, net asset flows for Canadian-domiciled sustainable and responsible investment mutual funds and ETFs reached CAD 5.3 billion in the first quarter of 2021, surpassing the CAD 3.3 billion recorded in 2020. This upward trend indicates a growing interest in and commitment to sustainable investing in the Canadian market.

Additionally, 17 sustainable and responsible investment funds debuted in the first quarter of 2021, compared to 40 in 2020. The interest in sustainable and responsible investing is still high, and the majority of retail investors want their financial services provider to inform them about available sustainable and responsible investment options, according to the 2020 RIA Investor Opinion Survey [66], which tracks individual investor's perspectives on responsible investing over time. Ultimately, 72% of the respondents in 2020

and 2019 expressed "very" or "somewhat" interest in responsible investment compared with 60% of the respondents in 2018. Nevertheless, only approximately 25% of retail investors in the poll had questions about responsible investing.

## 3. Materials and Methods

### 3.1. Methodology and Data

This study explored the connection between shareholder types and the use of ESG supply chain indicators. This study used an empirical methodology to analyze ESG and financial data from Refinitiv databases, which provided information on company characteristics, ownership, ESG disclosure, and financial performance. This research focused on the period from 2017 to 2022, during which ESG data were most readily available. The sample of listed companies was carefully selected to ensure that all necessary financial and ownership variables were available. Ultimately, this study included 2811 companies from various industries and locations around the world with different sizes. The sample size was large enough to include NAIC's sector classification, which helped to reduce the risk of industry bias in the results. Additionally, the sample size helped to mitigate the risk of company size bias in environmental data, as larger companies are often more likely to implement socially responsible programs [30,67]. Overall, the methodology and research design were well-justified and provided valuable insights into the relationship between shareholder types and ESG supply chain indicators.

### 3.2. Variables

ESG disclosure information was used as the dependent variable, with a particular emphasis on the environmental component. The Refinitiv database provides a comprehensive collection of over 150 company-level environmental metrics derived from publicly reported information. From this extensive dataset, this study carefully selected five key indicators relevant to supply chain performance, enabling an assessment of the presence or absence of disclosure. These selected measures were transformed into binary variables, which formed the basis for calculating the environmental supply chain index (ESCI). Table 1 presents the chosen measures along with their explanations from Refinitiv [68].

**Table 1.** Explanations of selected measured.

| Measure | Abbreviation | Description |
| --- | --- | --- |
| Environmental Materials Sourcing | EMS | Does the company claim to use environmental criteria (e.g., life cycle assessment) to source or eliminate material? |
| Environmental Supply Chain Management | ESCM | Does the company use environmental criteria (ISO 14000 [69]—energy consumption, etc.) in the selection process for its suppliers or sourcing partners? The data can also be on existing suppliers who were selected using some environmental criteria. |
| Policy Emissions | PE | Does the company have a policy to reduce emissions? This includes land, air, and water emissions stemming from the company's core activities- processes, mechanisms, or programs in place to reduce emissions in its operations system or a set of formal, documented processes for controlling emissions and driving sustained improvement. |
| Policy Environmental Supply Chain | PESC | Does the company have a policy to include its supply chain in the company's efforts to lessen its overall environmental impact? This includes legal compliance data on the supply chain to reduce environmental impact is in scope; data on collaboration with suppliers towards reducing their environmental impacts; and data on reducing environmental impacts of the supplier's operations. |
| Resource Reduction Policy | RRP | Does the company have a policy for reducing the use of natural resources or lessening the environmental impact of its supply chain? |

Source: Own elaboration.

### 3.3. Explanatory and Control Variables

Company financial performance data from Refinitiv was acquired and organized into three distinct groups. The first group comprised shareholder ownership variables, measured as the percentage of shares held by different types of investors according to the Refinitiv classification. Minor adjustments were made to the investor types; specifically, the "Investment Advisor/Hedge Fund" category was consolidated into "Investment Advisor", while "Hedge Fund Portfolio" and "Pension Fund Portfolio" were merged with "Hedge Fund" and "Pension Fund", respectively. Moreover, this study focused on investor types that had invested in at least 50% of the companies in the sample, resulting in seven investor types defined as explanatory variables and forming the primary focus of this study.

In addition to the shareholder ownership variables, financial performance variables were incorporated as control variables to mitigate potential bias effects. Three financial characteristics, namely, size, profitability, and financial leverage, were defined to control for such biases. Consequently, this study included "Company Market Capitalization" as a size variable, given its significance, as previous research has demonstrated that company size bias can positively affect environmental ratings, favoring larger capital companies [69,70]. To control for profitability and financial leverage, "Return on Assets" (ROA) and "Total Debt Percentage to Total Equity" were included, respectively.

Furthermore, a third group of variables was integrated, incorporating the NAIC's sector classification to address a potential bias associated with industry regulations. Only representative sectors that did not introduce multicollinearity issues into the model were considered. A comprehensive list of variables used in this study is presented in Table 2.

**Table 2.** Variables used in this study.

| Variable | Definition |
| --- | --- |
| Company Market Capitalization | The market value of outstanding shares |
| Total Debt Percentage to Total Equity | Percentage of debt over equity |
| ROA | Return on assets |
| Corporation | Percentage of shares held by corporations |
| Individual Investor | Percentage of shares held by individual investors |
| Pension Fund | Percentage of shares held by pension funds |
| Sovereign Wealth Fund | Percentage of shares held by sovereign wealth funds |
| Hedge Fund | Percentage of shares held by hedge funds |
| Insurance Company | Percentage of shares held by insurance companies |
| Investment Advisor | Percentage of shares held by investment advisors |
| Industry dummies | For each industry, there exists a binary variable that takes a value of 1 when the company belongs to the industry according to the NAIC classification. The industries considered are the following: finance and insurance; construction; information; quarrying and oil and gas extraction; professional, scientific, and technical services; real estate and rental and leasing; retail trade; transportation and warehousing; and wholesale and utilities. |

Source: Own elaboration.

Several transformations were applied to the variables in this study. First, the variable 'Company Market Capitalization' underwent a logarithmic transformation followed by standardization. Percentage variables were rescaled to a range of 0 to 100, while 'Return on Assets' (ROA) was multiplied by 100.

We investigated the potential strong and exogenous impact of the pandemic on the behavior of control and explanatory variables using Figures 1 and 2, which display their distribution over the years. The analysis reveals that the financial performance variables exhibit no significant changes across the years, indicating a likely homogeneous distribution throughout the period. Similarly, the ownership variables demonstrate consistent distribution patterns with non-substantial variations for any type of investor.

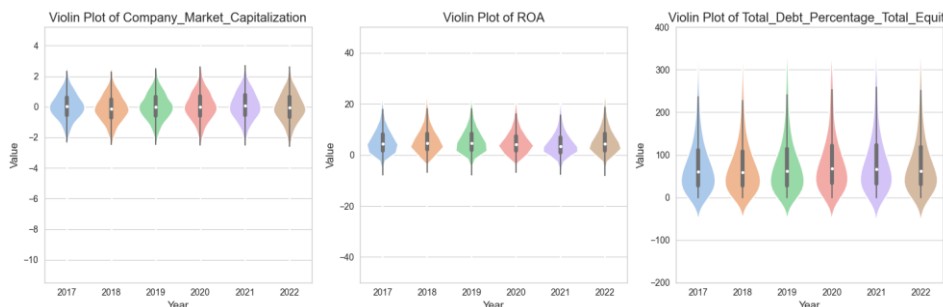

**Figure 1.** Distribution of financial performance variables from 2017 to 2022. Source: Own elaboration.

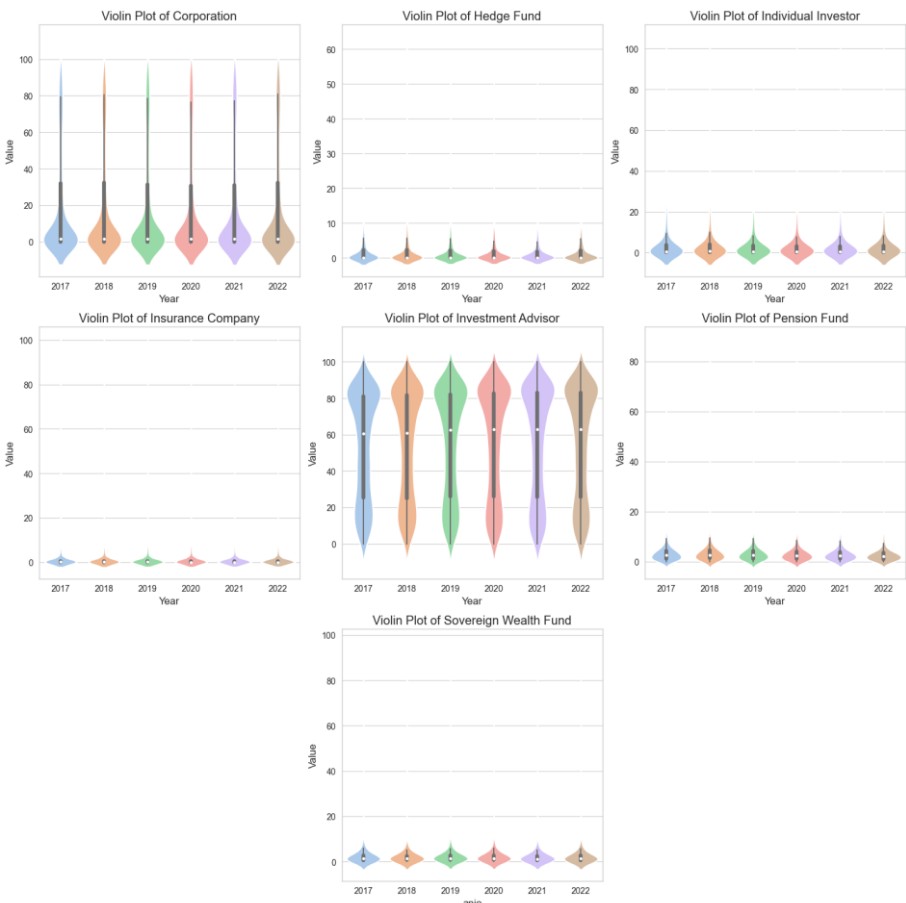

**Figure 2.** Distribution of ownership variables from 2017 to 2022. Source: Own elaboration.

### 3.4. Environmental Supply Chain Index (ESCI)

Previous research by Lai and Wong [71] and Liu et al. [72] investigated the relationship between environmental supply chain practices and corporate financial performance. However, further investigation is necessary to understand how these practices are linked to shareholder ownership. For this purpose, we identified 17 environmental supply chain disclosures or indicators in the Refinitiv ESG data, which allowed us to examine the environmental supply chain practices adopted by businesses. To address correlations among some of these indicators, we selected five key indicators for the analysis: environmental materials sourcing, environmental supply chain management, policy emissions, policy environmental supply chain, and resource reduction policy.

Following Wang and Sarkis [5], ref [64] and D'amato et al. [73], the environmental supply chain index (ESCI) is proposed as a measurement tool to evaluate the environmental performance of a company's supply chain. This index comprises five indicators that are equally weighted and range from 0 to 5. A score of 0 indicates that the company has not ful-

filled any of the indicators and has a low level of environmental supply chain achievement. On the other hand, a score of 5 indicates that the company has fulfilled all five indicators and has a high level of achievement. This index is treated as an ordered categorical variable since the indicators are binary in nature, similar to Likert-type variables [74]. An explanation of the indicators is provided in Table 1.

$$ESCI_{ti} = EMS_{ti} + ESCM_{ti} + PE_{ti} + PESC_{ti} + RRP_{ti}$$

where $i = 1, \ldots, 2811$ is the number of the company and $t$ = the year of observation.

The index was thoroughly validated with input from ten experts in environmental supply chain management. These experts consisted of five highly experienced academics who have spent over a decade working in the field of environmental supply chain management, as well as five seasoned practitioners who hold senior managerial positions within the environmental supply chain industry.

The behavior of the environmental supply chain index changes over time (Figure 3). There is a trend in which companies have a broader appropriation of environmental supply chain practices over the years. In this situation, there is an increase of 33% in category 4 and 51% in category 5 from 2017 to 2022 and a decrease of 77% in category 0 and 48% in category 1. Categories 2 and 3 are stable across these years. This suggests relevant movement in specific companies that may not be correlated with the control and explanatory variables.

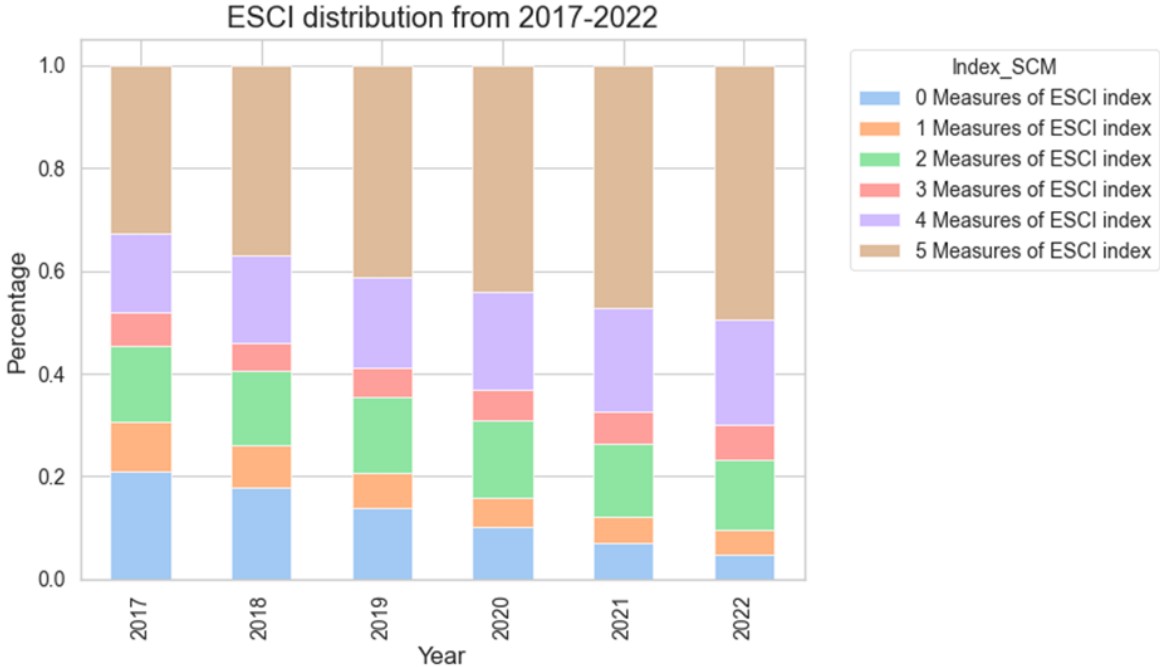

**Figure 3.** ESCI distribution from 2017–2022. Source: Own elaboration.

### 3.5. Model

The results of the environmental supply chain index show that each of the categories is representative in each year despite concentrations of higher values (Figure 3). However, observations for the same companies may be correlated and have relevant variations in ESCI values over the years. Hence, this study uses random effects ordered logistic regression to analyze these panel data with unobserved variations, considering the results of the distributions' behavior over the years for ESCI and control and ownership variables. This type of ordered logistic regression model permits analyzing the probability of each company belonging to a specific category depending on the explained and control variables [75] and extends the analysis by introducing a random variable that resolves unobserved variation in companies.

The model considers the following equation, where *j* represents the number of categories in the ordinal variable, $i = 1, \ldots, 2811$ represents the number of the company, *t* represents the year of observation, $X_i$ represents the explanatory and control variables, $\beta_j$ represents the estimated coefficient for each explanatory and control variable, and $u_j$ represents the random effect, which is unobserved and usually assumed to vary among individuals according to a normal distribution [76].

$$Pr(Y_{it} > j) = \frac{\exp\left(\alpha_j + X_i\beta_j + u_j\right)}{1 - \exp\left(\alpha_j + X_i\beta_j + u_j\right)}, j = 0, 1, 2, 3, 4, 5$$

## 4. Results

Table 3 presents the results of the random effects ordered logistic regression. Wald tests on the set of variables indicate that the coefficients are significantly not jointly equal to zero (Fstat. = 736.78, $p\backslash0.000$). This means that the model's fit is statistically significant. The signs of the estimated coefficients are interpreted as the ordered log odds of the ESCI. A positive coefficient suggests that an increase in the explanatory variable is associated with a higher likelihood of moving to a higher category of the ESCI. Conversely, a negative coefficient indicates a decrease in the likelihood of moving to a higher category [77].

**Table 3.** Random effects ordered logistic regression.

| | Random Effects Ordered Logistic Regression | |
| --- | --- | --- |
| | Coef. | *p* > |z| |
| Company Market Capitalization | 1.416275 | 0.000 *** |
| ROA | −0.0251847 | 0.000 *** |
| Total Debt Percentage to Total Equity | 0.0000531 | 0.000 *** |
| Finance and Insurance | −4.236074 | 0.000 *** |
| Construction | −1.263819 | 0.004 *** |
| Information | −2.832346 | 0.000 *** |
| Quarrying and Oil and Gas Extraction | −2.302227 | 0.000 *** |
| Professional, Scientific, and Technical Services | −2.589448 | 0.000 *** |
| Real Estate, Rental, and Leasing | −2.442437 | 0.000 *** |
| Retail Trade | −1.593914 | 0.000 *** |
| Transportation and Warehousing | −2.36935 | 0.000 *** |
| Utilities | −0.83461 | 0.019 ** |
| Wholesale | −1.987017 | 0.000 *** |
| Corporation | −0.0067893 | 0.132 |
| Individual Investor | −0.0285433 | 0.000 *** |
| Pension Fund | 0.0052381 | 0.649 |
| Sovereign Wealth Fund | 0.0217324 | 0.031 ** |
| Hedge Fund | −0.0542998 | 0.000 *** |
| Insurance Company | 0.0197999 | 0.149 |
| Investment Advisor | −0.0145407 | 0.001 *** |
| sigma2_u | 17.66762 | |

Source: Own elaboration. (1) *** Statistically significant 99% C.I. level; ** statistically significant 95% C.I. level; (2) Threshold coefficients: 0|1: −8.972023; 1|2: −7.633363; 2|3: −5.38196; 3|4: −4.497614; and 4|5: −1.943955.

To analyze the magnitude of the explanatory variables, marginal effects were calculated and shown in Table 4. Given that the ownership variables are on a scale from 0 to 100 percent, the coefficients from the marginal effects can be interpreted as follows. The coefficient is the marginal probability for a determined category given a 1% increase in the share percentage held by a determined investor type. Therefore, these results allow for an analysis of the magnitude and direction of the relation between ESCI and ownership variables.

**Table 4.** Marginal effects.

| | Category 0 | | Category 1 | | Category 2 | | Category 3 | | Category 4 | | Category 5 | |
|---|---|---|---|---|---|---|---|---|---|---|---|---|
| | Coef. | $p > \lvert z \rvert$ | Coef. | $p > \lvert z \rvert$ | Coef. | $p > \lvert z \rvert$ | Coef. | $p > \lvert z \rvert$ | Coef. | $p > \lvert z \rvert$ | Coef. | $p > \lvert z \rvert$ |
| Company Market Capitalization | −0.0532 | 0.000 *** | −0.01707 | 0.000 *** | −0.02624 | 0.000 *** | −0.00775 | 0.000 *** | −0.00458 | 0.000 *** | 0.10892 | 0.000 *** |
| ROA | 0.00094 | 0.000 *** | 0.00030 | 0.000 *** | 0.00046 | 0.000 *** | 0.00013 | 0.000 *** | 0.00008 | 0.004 *** | −0.00193 | 0.000 *** |
| Total Debt Percentage to Total equity | $-2.00 \times 10^{-6}$ | 0.000 *** | $-6.41 \times 10^{-7}$ | 0.000 *** | $-9.84 \times 10^{-7}$ | 0.000 *** | $-2.91 \times 10^{-7}$ | 0.000 *** | $-1.72 \times 10^{-7}$ | 0.022 ** | $4.09 \times 10^{-6}$ | 0.000 *** |
| Finance and Insurance | 0.15928 | 0.000 *** | 0.05108 | 0.000 *** | 0.07850 | 0.000 *** | 0.02320 | 0.000 *** | 0.01371 | 0.000 *** | −0.32579 | 0.000 *** |
| Construction | 0.04752 | 0.004 *** | 0.01523 | 0.004 *** | 0.02342 | 0.004 *** | 0.0069 | 0.002 *** | 0.00409 | 0.022 ** | −0.09720 | 0.004 *** |
| Information | 0.10650 | 0.000 *** | 0.03415 | 0.000 *** | 0.05249 | 0.000 *** | 0.01551 | 0.000 *** | 0.00917 | 0.000 *** | −0.21783 | 0.000 *** |
| Quarrying and Oil and Gas Extraction | 0.08656 | 0.000 *** | 0.02776 | 0.000 *** | 0.04266 | 0.000 *** | 0.01261 | 0.000 *** | 0.00745 | 0.001 *** | −0.17706 | 0.000 *** |
| Professional, Scientific, and Technical Services | 0.09736 | 0.000 *** | 0.03122 | 0.000 *** | 0.04799 | 0.000 *** | 0.01418 | 0.000 *** | 0.00838 | 0.001 *** | −0.19915 | 0.000 *** |
| Real Estate, Rental, and Leasing | 0.09184 | 0.000 *** | 0.02945 | 0.000 *** | 0.04526 | 0.000 *** | 0.01338 | 0.000 *** | 0.00790 | 0.001 *** | −0.18784 | 0.000 *** |
| Retail Trade | 0.05993 | 0.000 *** | 0.01922 | 0.000 *** | 0.02954 | 0.000 *** | 0.00873 | 0.000 *** | 0.00516 | 0.006 *** | −0.12258 | 0.000 *** |
| Transportation and Warehousing | 0.08909 | 0.000 *** | 0.02857 | 0.000 *** | 0.04391 | 0.000 *** | 0.01297 | 0.000 *** | 0.00767 | 0.001 *** | −0.18222 | 0.000 *** |
| Utilities | 0.03138 | 0.019 ** | 0.01006 | 0.019 ** | 0.01546 | 0.021 ** | 0.00457 | 0.019 ** | 0.00270 | 0.042 ** | −0.06418 | 0.019 ** |
| Wholesale | 0.07471 | 0.000 *** | 0.02396 | 0.000 *** | 0.03682 | 0.000 *** | 0.01088 | 0.000 *** | 0.00643 | 0.003 *** | −0.15282 | 0.000 *** |
| Corporation | 0.00025 | 0.133 | 0.00008 | 0.133 | 0.00012 | 0.134 | 0.00003 | 0.138 | 0.00002 | 0.147 | −0.00052 | 0.133 |
| Individual Investor | 0.00107 | 0.000 *** | 0.00034 | 0.000 *** | 0.00052 | 0.000 *** | 0.00015 | 0.000 *** | 0.00009 | 0.002 *** | −0.00219 | 0.000 *** |
| Pension Fund | −0.0001 | 0.649 | −0.00006 | 0.650 | −0.00009 | 0.650 | −0.00002 | 0.649 | −0.00001 | 0.652 | 0.00040 | 0.649 |
| Sovereign Wealth Fund | −0.0008 | 0.031 ** | −0.00026 | 0.033 ** | −0.00040 | 0.032 ** | −0.00011 | 0.032 ** | −0.00007 | 0.054 * | 0.00167 | 0.031 ** |
| Hedge Fund | 0.00204 | 0.000 *** | 0.00065 | 0.000 *** | 0.00100 | 0.000 *** | 0.00029 | 0.000 *** | 0.00017 | 0.004 *** | −0.00417 | 0.000 *** |
| Insurance Company | −0.0007 | 0.149 | −0.00023 | 0.151 | −0.00036 | 0.150 | −0.00010 | 0.148 | −0.00006 | 0.176 | 0.00152 | 0.149 |
| Investment Advisor | 0.00054 | 0.001 *** | 0.00017 | 0.001 *** | 0.00026 | 0.001 *** | 0.00007 | 0.001 *** | 0.00004 | 0.008 *** | −0.00111 | 0.001 *** |
| Predicted outcomes | Pr (ESCI == 0) (predict, outcome(0)) 0.1165331 | | Pr (ESCI == 1) (predict, outcome(1)) 0.0583815 | | Pr (ESCI == 2) (predict, outcome(2)) 0.1334956 | | Pr (ESCI == 3) (predict, outcome(3)) 0.0628442 | | Pr (ESCI == 4) (predict, outcome(4)) 0.1968036 | | Pr (ESCI == 5) (predict, outcome(5)) 0.4319419 | |

Source: Own elaboration. *** Statistically significant 99% C.I. level; ** statistically significant 95% C.I. level.

## 4.1. Financial Variables

The inclusion of Company Market Capitalization helped to control the size bias on the environmental rating presented [69,70], showing that higher values of this variable increase the probability of being in the superior categories. In Table 4, the results for the marginal effects show that the coefficient of each category is statistically significant at 99% confidence, and the probabilities are higher in the extreme categories, considering its direction. Company Market Capitalization has a marginal probability of 10% for being in category 5, which decreases as we move down the categories until category 0, where the marginal probability of being in that category is −5%.

The control variable "Total Debt Percentage to Total Equity", which represents leverage, exhibits a similar pattern of results but with a significantly lower magnitude. The marginal effect results indicate that the effect of each category is statistically significant with a confidence level of 99%. Additionally, for every increase of $4.09 \times 10^{-6}$ in "Total Debt Percentage to Total Equity", the marginal probability of being in the superior category increases, while the probability of being in the remaining categories decreases.

In the case of the profitability variable "ROA Margin", it exhibits a contrasting result compared with "Market Capitalization" and "Total Debt Percentage to Total Equity." "ROA Margin" is also statistically significant at a 99% confidence level across all categories. However, a 1% increase in "ROA Margin" will lead to a decrease of −0.1% in the probability of being in category 5. Conversely, a 1% increase in "ROA Margin" will increase the probability of being in the other categories. The lower categories show larger effects, with category 0 having the highest magnitude, resulting in a probability increase of 0.09% with a 1% increase in "ROA Margin".

### 4.2. Industry Variables

The focus of the empirical analysis results is on determining whether the explanatory factors have positive or negative impacts. In this regard, the findings demonstrate that the use of control variables lowers the likelihood of analysis-related bias, such as bias based on the sector to which the company belongs. The majority of industry variables have a considerable impact on each category and exhibit similar behavior.

This behavior implies that an increment would increase the probability of companies belonging to the lower categories while decreasing the probability of being in the superior categories. For instance, the construction industry exhibits a 4% marginal probability of being in category 0 and a −9% marginal probability of being in category 5; the information industry shows a 10% marginal probability of being in category 0 and a −21% marginal probability of being in category 5; the professional, scientific, and technical services industry demonstrates a 9% marginal probability of being in category 0 and a −19% marginal probability of being in category 5; the real estate, rental, and leasing industry has a 9% marginal probability of being in category 0 and a −18% marginal probability of being in category 5; the quarrying and oil and gas extraction industry exhibits an 8% marginal probability of being in category 0 and a −17% marginal probability of being in category 5; the retail trade industry shows a 5% marginal probability of being in category 0 and a −12% marginal probability of being in category 5; the transportation and warehousing industry demonstrates an 8% marginal probability of being in category 0 and a −18% marginal probability of being in category 5; the utilities industry has a 3% marginal probability of being in category 0 and a −6% marginal probability of being in category 5; the finance and insurance industry shows a 15% marginal probability of being in category 0 and a −3% marginal probability of being in category 5; and finally, the wholesale industry exhibits a 7% marginal probability of being in category 0 and a −15% marginal probability of being in category 5. These results indicate that the industry to which a company belongs complements the variable Company Market Capitalization, thereby correcting the effect of size bias.

### 4.3. Ownership Variables

The results indicate that the relationship between stakeholder ownership and ESCI can be categorized into three groups. The first group comprises hedge funds, individual investors, and investment advisors. Higher percentages of stocks owned by these investor types are associated with an increased probability of falling into the inferior categories of the ESCI, while the probability of being in the superior categories decreases. Hedge funds show statistical significance in each category, with positive and substantial probabilities of 0.2% and 0.1% for being in categories 0 and 2, respectively, and a negative impact of −0.4% for being in category 5. Similarly, individual investors exhibit a similar pattern, being statistically significant in each category, with a notable probability of 0.01% for being in category 0 and a negative marginal probability of −0.2% for being in category 5. In the case of investment advisors, their influence is of lower magnitude compared to the previous investor types. They demonstrate a marginal probability of 0.05% for being in category 0 and −0.1% for being in category 5.

On the contrary, sovereign wealth funds exhibit a positive association with environmental supply chain performance, wherein a higher percentage of stocks owned by this investor type increases the probability of belonging to superior categories of the ESCI and reduces the probability of falling into inferior categories. Specifically, for being in category 5, sovereign wealth funds have a marginal probability increase of 0.1% for each share percentage held, while membership in the remaining categories experiences a decrease in marginal probability.

Finally, the third group has no statistical relationship with any category, this group includes investor types such as corporations, insurance companies, and pension funds. However, there is evidence in the literature that these types of investors have a strong relationship in their investment behavior with regional regulations. Therefore, there is an

opportunity for future research to concentrate on how this regulation affects the investment interest of this type of investor in ESG and environmental supply chain practices.

## 5. Discussion

### 5.1. Long-Term Horizon Investments Correlate with Greater Use of ESG

Based on the analysis conducted, there is a positive relationship between long-term-oriented investors, such as sovereign wealth funds, pension funds, and insurance companies, and sustainable performance, as measured by the adoption of ESG practices by companies and supply chains. These investors demonstrate a more comprehensive understanding of the businesses they support, which can lead to reduced risk exposure and identification of potential growth opportunities. Consequently, sustainable practices have become a priority in their investment decisions, gaining increasing importance and shaping a future focused on overall sustainability. This aligns with the findings of Carter & Rogers [29], who observed that companies adopting sustainable practices in supply chain management tend to be market leaders and reap long-term economic benefits. It also corresponds with Aldowaish et al.'s [45] research, which highlights how incorporating ESG factors into business operations enhances sustainability, making the company more attractive to investors. Furthermore, institutional investors are placing greater emphasis on environmental issues when making investment decisions [20].

In recent times, there has been a significant shift in perspective toward sustainable practices in the business world. Previously, investors placed greater value on physical assets such as property, equipment, and machinery. However, today, the worth of companies is largely attributed to intangible factors like reputation, corporate culture, and customer loyalty [78]. The traditional focus on short-term profit maximization, based solely on shareholder value, is now transforming to incorporate ethical and sustainable operations, ensuring a stable position in the economy over the long term. Consequently, sustainability and long-term value preservation for shareholders are gaining increased attention. Moreover, the public perception of corporations has evolved from merely being financial market players to entities with a responsibility to contribute positively to society and the environment [78]. The apparent trade-off between short-term returns and long-term value is diminishing, as companies with higher sustainability not only demonstrate superior environmental and social performance but also deliver higher expected returns for their owners, indicating that doing good pays off [78,79].

Considering long-term horizon investment, the incorporation of environmental, social, and governance (ESG) elements becomes crucial for several reasons. Firstly, systematically integrating ESG factors serves as a risk management strategy, particularly in relation to potential financial impacts that may arise over extended periods due to non-compliance with sustainability standards, leading to fines or penalties. Secondly, regulators increasingly recognize the importance of ESG factors as part of their fiduciary duty to investment managers. Additionally, investors are placing greater emphasis on transparency, demanding a clear understanding of how their investments affect individuals, communities, and the world at large. Furthermore, there is growing evidence supporting the idea that financial sustainability goes beyond traditional measures and needs to encompass a broader range of external factors. By doing so, its aim is to maximize long-term returns and profits while mitigating controversies that could erode stakeholders' trust. A tangible illustration of the significance of ESG can be found in the substantial adoption of ESG strategies worldwide. Approximately USD 20 trillion, about one-fourth of all professionally managed assets globally, is now invested based on ESG principles. Moreover, over 2500 signatories, managing over USD 80 trillion in assets, have committed to adhere to the United Nations Principles for Responsible Investment [80].

According to Bellandi [49], firms are increasingly recognizing the importance of striking a balance between their financial goals and sustainable development using ESG measures. This enables companies to establish a balanced portfolio of projects that promote sustainable growth and long-term value while maintaining consistency between financial

and ESG growth objectives. Over a long-time horizon, ESG risks are more likely to surface, and unsustainable practices are likely to incur costs, negatively impacting the financial returns of companies that poorly manage ESG risks. Therefore, it is not merely a passing trend to consider ESG performance more significantly in financial decision-making; rather, it represents a larger redefinition of the role of long-term investments in businesses. Corporations are moving away from exclusive shareholder value maximization and embracing broader sustainability aspects [80].

*5.2. Short-Term Horizon Investments Could Translate into Poor ESG Adoption*

As derived from the results, investment sources with a shorter-term horizon showed a negative correlation with ESG indicator adoption within companies and their supply chains. The reasoning behind these results could rest in factors that investors face in their investment decisions such as conflicting priorities, lack of incentives, and reduced engagement or influence. Considering that short-term horizon investors are primarily focused on maximizing short-term financial gains, often driven by quarterly or annual performance metrics, the fact that ESG considerations often require long-term investments and commitments presents conflicting priorities, as these investments may have longer payback periods or require additional resources, which short-term investors may not be willing to allocate due to their focus on immediate profits [2].

Moreover, the pressure to deliver immediate results can create a misalignment between the time horizon of short-term investor decisions and the longer-term benefits associated with ESG practices. ESG adoption often yields positive outcomes over the long run, both in terms of financial performance and the broader impact on the environment and society. However, short-term investors may not see sufficient incentives to prioritize ESG adoption since their performance metrics are typically focused on shorter timeframes [2]. On top of that, short-term investors' reduced engagement and limited holding periods hinder their ability to actively engage with companies and advocate for ESG improvements. ESG adoption often requires active engagement with companies to encourage positive change in their practices. However, short-term investors, who tend to hold investments for shorter periods, have limited opportunities to influence companies positively and drive meaningful ESG changes.

Contrary to much of the existing literature, Gibson Brandom et al. [81] found that Morningstar Sustainability Ratings did not indicate any outperformance of high sustainability-rated mutual funds compared with the lowest-rated funds. Despite the top-rated funds attracting more capital, they did not outperform the lowest-rated ones. Similarly, Raghunandan and Rajgopal [82] compared the ESG performance of companies in ESG fund portfolios with non-ESG portfolios. They discovered that companies in ESG portfolios had a weaker track record of adhering to labor and environmental regulations and adding companies to ESG portfolios did not lead to an improvement in labor or environmental compliance [83]. Consequently, short-term investors may prefer to prioritize immediate gains over the possibility of larger long-term returns, which could lead them to focus on companies with lower ESG emphasis and potentially poorer sustainable performance in the long run, which not always leads to higher profits.

*5.3. Hedge Funds Case with ESG Adoption*

In general, the findings of this study indicate that as hedge fund investment increases, the likelihood of a company being in the higher categories of the ESG index decreases, while the probability of being in the lower categories increases. There is a trend indicating a growing acceptance of ESG in asset management across various sectors, including alternative investments like hedge funds. However, compared to other institutional investors, the hedge fund industry has been relatively slower at adopting ESG practices [84]. The diverse nature of their strategies, the types of assets they trade, and their potential for short selling could potentially hinder their full commitment to ESG principles. These factors

may explain why the results for hedge funds did not align with the expected trend in ESG adoption in their investments.

Consistent with this view, as mentioned before, Liang et al. [44] state that hedge funds are a crucial component of the portfolios of institutional investors who have adopted responsible investing, but given their lack of transparency, disclosure, and regulatory oversight as well as the divergence in opinions between investors and fund managers regarding the significance of ESG, they are particularly vulnerable to detrimental procedures such as greenwashing. Additionally, the intricate tactics utilized by hedge funds, coupled with their limited transparency, disclosure, and regulatory supervision, heighten the likelihood of agency issues and opportunistic actions. This indicates a departure from the anticipated norms of business conduct in terms of compliance, aligning with the principles of agency theory [44].

These authors add find that hedge fund managers, in contrast with other alternative investment managers, are substantially more skeptical than their investors about the importance of ESG [44]. Additionally, "relative to mutual funds, the complex strategies employed by hedge funds and their lower levels of transparency standards amplify the potential for agency problems and opportunistic behavior" [44] (p. 1586). Similarly, another reason for the low level of ESG adoption among hedge fund managers is that it is impossible to apply ESG factors to hedge fund strategies, such as systematic trend-following and discretionary global macro, because these approaches are top-down and reliant on market conditions [85].

It is important to recognize that hedge funds may have varying investment strategies and time horizons. Some engage in frequent trading with shorter investment horizons, while others adopt longer-term investment approaches, holding assets for extended periods. On the other hand, institutional investors like pension funds and sovereign wealth funds typically have longer-term investment horizons, driven by their obligations to provide financial security or support specific initiatives over extended periods. Long-term investors often prioritize stability, risk management, and sustainable growth, making them more aligned with the integration of ESG practices. While there might be hedge funds that incorporate ESG considerations into their strategies, the overall correlation between hedge funds and ESG practices is not as strong as that of long-term institutional investors. This study supports the notion that the nature of hedge funds differs from other institutional investments, leading to different results.

### 5.4. Policy Implications

The policy implications of these results for both businesses and governments are significant. Governments can play a crucial role in promoting sustainable practices among long-term investors. They can provide incentives such as tax breaks and cuts for investors who prioritize sustainability, thus encouraging a long-term investment perspective. Additionally, governments can implement regulations requiring institutional investors to integrate ESG considerations into their investment plans. To further encourage sustainable investment practices, they can establish frameworks for sustainable finance and collaborate with financial institutions to provide educational resources and programs. Strengthening reporting regulations can ensure that businesses disclose their ESG practices and performance, enhancing transparency standards and enabling investors to make informed decisions. Governments can also coordinate policies with international frameworks and participate in global initiatives to promote sustainable practices on a global scale. Using these measures, governments can foster a more sustainable investment landscape and drive positive change toward sustainability.

On the other hand, for the business sector, there are several policy implications that can help promote sustainable practices among investors. Businesses can revise their corporate governance structures to ensure the inclusion of sustainability considerations and set long-term sustainability goals aligned with ESG factors. Engaging with long-term investors and stakeholders to understand their sustainability expectations and fostering collaboration

for sustainable practices is essential. Moreover, enhanced ESG disclosure and reporting practices can provide transparent information to long-term investors, facilitating informed decision-making. Integrating sustainability into strategic decision-making processes and collaborating with sustainable investors can further align business strategies and practices with the expectations of long-term investors. Finally, prioritizing research and development efforts for sustainability-oriented innovation can attract long-term investors who seek environmental and social impact opportunities and create aggregate value in the long run.

These policy implications collectively contribute to the promotion of sustainable practices among investors, with governments and businesses working together to create and enable an environment that drives the integration of sustainability into investment decisions. Therefore, both governments and corporations may promote a more sustainable financial system, create long-term wealth, and have a good influence on the environment and society by putting these measures into place.

## 6. Conclusions

This paper investigates how different investors consider ESG factors in their investment decisions and whether company ownership impacts environmental performance. This study uses data from the Refinitiv database and categorical logit regression analysis. The results indicate that companies financed with long-term investment sources demonstrate a higher degree of environmental sustainability indicators in their supply chain compared to those funded with short-term sources. Specifically, long-term investors like sovereign wealth funds, pension funds, and insurance companies show a strong and favorable association with the adoption of ESG variables in the supply chain. These investors prioritize sustainability factors when making decisions, valuing sustainable supply chain practices in their investments. On the other hand, short-term investment sources exhibit a negative link with ESG practices, as they prioritize revenue over sustainability. These findings suggest that policymakers and private companies interested in enhancing sustainability should focus on longer-term sources of investment, as they are more committed to ESG practices and are more likely to adopt environmental practices in the supply chain. Additionally, it is worth noting that companies with higher market capitalization tend to have a broader adoption of sustainable supply chain practices, regardless of ownership composition.

The COVID-19 pandemic in 2020 did not cause an alteration in the behavior of the studied variables. We found that the financial performance variables exhibit no significant changes across the years 2017–2022, indicating a likely homogeneous distribution throughout the period. Similarly, the ownership variables demonstrate consistent distribution patterns with non-substantial variations in any type of investor.

These findings have significant implications for both academic research and practical applications. In terms of academic research, this paper contributes to the growing body of literature on corporate environmental responsibility by providing empirical evidence of how ownership structures influence environmental practices in supply chains. The results highlight the importance of considering ownership characteristics as key determinants of environmental performance in future research endeavors.

From a practical standpoint, the insights from this study offer valuable guidance to companies and policymakers seeking to enhance environmental sustainability within supply chains. Understanding the impact of ownership on environmental practices can aid in the development and implementation of targeted initiatives and policies that promote responsible and sustainable behavior. Encouraging ownership structures that prioritize long-term perspectives and stakeholder engagement can foster a culture of environmental stewardship throughout the supply chain.

However, it is essential to recognize that the relationship between ownership and environmental practices is complex and can be influenced by various contextual factors. Future research should continue to explore additional dimensions of ownership, such as government ownership, and their impact on environmental practices. Moreover, conduct-

ing comparative studies across different industries and regions can provide further insights into the generalizability and robustness of the findings.

This study provides valuable insights into the influence of equity ownership on ESG practice disclosures in the supply chain, underscoring the significance of considering long-term funding sources to promote sustainability and environmentally friendly company practices. Additionally, this study contributes to the existing literature by shedding light on how different equity funding sources impact ESG adoption in the supply chain's environmental aspect. The findings highlight the importance of considering the investment horizon and priorities of various types of equity investors while developing sustainable investment strategies. By enhancing our understanding of the relationship between equity funding and supply chain sustainability, this study can assist companies and investors in making more informed decisions and encouraging the adoption of sustainable practices in global supply chains.

*Future Research*

To advance our understanding of the relationship between investment and ownership types and sustainable supply chain practices, several areas warrant further research. For instance, understanding the motives behind the priority of sustainable practices by long-term investors and the demotion of ESG factors by investors with shorter-term views will need an emphasis on examining investor behavior and decision-making. Additionally, research should explore how different policy frameworks and regulations influence investor preferences and company behaviors concerning sustainable supply chain practices. Investigating the effectiveness of policy interventions and the need for further regulatory measures would contribute to understanding the role of external factors in shaping sustainable practices.

**Author Contributions:** Conceptualization: L.R., G.M., N.O. and I.P.-G.; methodology and validation: Moreno Gabriel, L.R. and N.O.; formal analysis: N.O. and G.M.; investigation: G.M., N.O., L.R. and I.P.-G.; data curation: G.M. and L.R.; writing—original draft: L.R., G.M., N.O. and I.P.-G.; writing—review and editing: L.R., N.O. and I.P.-G.; project administration: N.O. and L.R. All authors have read and agreed to the published version of the manuscript.

**Funding:** This research received no external funding.

**Institutional Review Board Statement:** The present study did not involve humans or animals.

**Informed Consent Statement:** Not applicable.

**Data Availability Statement:** The data used in this study were downloaded from Refinitiv [68] between 10 October 2022 and 21 July 2023.

**Conflicts of Interest:** The authors declare no conflict of interest.

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
