# Peer review of "The Effect of Company Ownership on the Environmental Practices in the Supply Chain: An Empirical Approach"

_sustainability, doi:10.3390/su151612450_

Round 1

Reviewer 1 Report

The submitted paper presents a very interesting ESG-based investigation of the relationship between company ownership and environmental practices, with a particular focus on environmental supply chain disclosures. The topic is of high interest for research and practice, the literature review and the thorough comparison between the study results and the findings of other research are the strengths of this paper.

However, there are some issues that need to be addressed and improved, in my opinion.

1.      Please pay closer attention to the in-text citations. I suggest that you avoid quoting paragraphs from the texts of other works (unless it is highly relevant to make a point) and that you paraphrase the ideas from the studies in question better. In my opinion, the ideas quoted from other works are excessive in the Introduction section and in section 5.1, and should simply be paraphrased. It is also difficult to follow the reflections of this paper and those of other studies because of the excessive and sometimes inappropriate use of quotation marks (for example, you open the quotes at line 107 and close them at line 110, but also at line 113).

2.      Also, please carefully check the in-text citations and the corresponding reference numbers. Under subsection 2.4, line 294, you inserted the citation of authors instead of reference numbers in brackets as indicated in the journal manual (Formentini & Taticchi, 2016) and the reference is not in the Reference list. On lines 216 and 427 you refer to a study conducted by Morgan Stanley in 2019, but with no corresponding reference in the Reference list. Similar situations are in lines 434 and 464.

3.      In Table 1, page 11, the description of abbreviation PESC repeats itself in two separate lines. Where do you use this abbreviation further in the study?

4.      Please explain the equation on line 534 (????? =???? + ????? +????+????+????). What does “i” mean? Please explain the abbreviations PEP and ESC (you can insert those descriptions in Table 1 and refer to this Table just before the equation on line 534). Also, the dependent variable in the equation is ECSI, whereas the abbreviation for the Environmental Supply Chain Index is ESCI (line 527) – which is correct?

5.      Please elaborate on how the 6 categories in Table 3 are defined.

6.  Check the explanations on lines 620-624. The probability of being in the lower categories : for Finance &Insurance category 5 is shown with a probability of 13% - is this correct?

Author Response

Reviewer 1:

  1. Please pay closer attention to the in-text citations. I suggest that you avoid quoting paragraphs from the texts of other works (unless it is highly relevant to make a point) and that you paraphrase the ideas from the studies in question better. In my opinion, the ideas quoted from other works are excessive in the Introduction section and in section 5.1, and should simply be paraphrased. It is also difficult to follow the reflections of this paper and those of other studies because of the excessive and sometimes inappropriate use of quotation marks (for example, you open the quotes at line 107 and close them at line 110, but also at line 113).

R: Thank you very much for your comment. We have already removed most quoted citations, particularly in the introduction and section 5.1, as we recognized that they provided only supplementary information. Consequently, we rephrased them as citations in a more concise manner.

  1. Also, please carefully check the in-text citations and the corresponding reference numbers. Under subsection 2.4, line 294, you inserted the citation of authors instead of reference numbers in brackets as indicated in the journal manual (Formentini & Taticchi,2016) and the reference is not in the Reference list. On lines 216 and 427 you refer to a study conducted by Morgan Stanley in 2019, but with no corresponding reference in the Reference list. Similar situations are in lines 434 and 464.

R: Thank you for your valuable comment. We had addressed those inconsistencies in our paper. In this revised version, we have incorporated the mentioned references. We also checked all the references to warrantee no more missing references.

  1. In Table 1, page 11, the description of abbreviation PESC repeats itself in two separate lines. Where do you use this abbreviation further in the study?

R: Thank you for your valuable comment; it has enabled us to address the inconsistencies in our paper. Specifically, we corrected all the abbreviations of the variables throughout the paper.

  1. Please explain the equation on line 534 (????? =???? + ????? +????+????+????). What does “i” mean? Please explain the abbreviations PEP and ESC (you can insert those descriptions in Table 1 and refer to this Table just before the equation on line 534). Also, the dependent variable in the equation is ECSI, whereas the abbreviation for the Environmental Supply Chain Index is ESCI (line 527) – which is correct?

R: Thank you for your comment. We expanded the description of the coefficients presented in our ESCI index. We also corrected the inconsistency of the abbreviation, the right one was ESCI = Environmental Supply Chain Index.

  1. Please elaborate on how the 6 categories in Table 3 are defined.

R: We sincerely appreciate your comment. It prompted us to replace the previous table with a new one that provides comprehensive explanations for each category of our ESCI index.

  1. Check the explanations on lines 620-624. The probability of being in the lower categories: for Finance &Insurance category 5 is shown with a probability of 13% - is this correct?

R: The analysis and explanation of the results have been updated with new findings obtained by incorporating data from additional years into our model. We are grateful for your comment, which led us to make these improvements. As a result, we have taken great care in providing a more detailed explanation of the probabilities associated with each category.

Reviewer 2 Report

Dear Authors,

Your paper addresses an interesting topic. Does ownership structure affect  ESG practices? You built an ESGI to measure ESG practices focusing on the environmental dimension and used Refinitiv data for 2021 to establish the relationship. In my opinion, there are several opportunities for improvement.

1. Given that you focus on the E of ESG, your abstract could be improved

2. Your Introduction lacks a synthesis of sample selection and contributions. 

3. You use data for a single year: the 2021 cross-section. This is too restrictive to reach generalizable conclusions. Moreover, temporal effects are impossible to account for. Adding more years to your research design is necessary.

4. Having your baseline results corroborated by some robustness tests would be interesting. For example, taking the sum or the average of your ESGI would allow for alternative estimation methods that would allow for easier testing of the effect of the ownership structure.

5. In its present form, your section 4.3 is oversimplified. For example, the effect of pension fund, sovereign wealth fund, and insurance company is always positive ...

Author Response

Reviewer 2 

  1. Given that you focus on the E of ESG, your abstract could be improved.

R: Thank you for your feedback; the abstract has been updated accordingly since our primary focus lies on the "E" aspect of the ESG indicators. Additionally, we included a brief justification for concentrating on the environmental aspect of ESG given that the environmental aspect is the one most directly related to supply chain activities and there is available related data for analysis.

  1. Your introduction lacks of synthesis of sample and contributions.

 R: Thanks for your comment. In this new version of the paper, we included a paragraph at the end of the introduction with summary of the sample and the contributions.

  1. You use data for a single year: the 2021 cross-section. This is too restrictive to reach generalizable conclusions. Moreover, temporal effects are impossible to account for. Adding more years to your research design is necessary.

 R: Thank you for this critical observation. We changed the model methodology to implement a data panel analysis from 2017 to 2022, instead of a cross-section analysis. We recognized the necessity to resolve the possible temporal effects presented in the data, especially considering that 2021 may have variations due to the pandemic. Thanks to your comment we were able to reach a more robust model.

This change is presented in sections “3. Materials and Methods” and “4. Results” on pages 10 to 19 from lines 483 to 713. A general overview, the new model “Random-effects ordered logistic regression” controls the company's random effects and there well added two new graphs show no relevant temporal effects in control and explanatory variables.

Figure 1. Distribution of financial performance variables from 2017 to 2022

Figura 1

Figure 2. Distribution of ownership variables from 2017 to 2022

Figura 2

  1. Having your baseline results corroborated by some robustness tests would be interesting. For example, taking the sum or the average of your ESGI would allow for alternative estimation methods that would allow for easier testing of the effect of the ownership structure.

 R: Thank you for your valuable and constructive comment. As you suggested, we have improved the methodology of our analysis by employing a data panel variation: Random-effects ordered logistic regression. This modification has greatly enhanced the robustness of our results. Although the ESCI variable is ordinal (non-continuous), we conducted the robustness test Wald Test, and the results demonstrated a statistically significant model fit (Fstat. = 736.78, p < 0.000). Given that and the rigorous statistical assessment, we don´t include additional robustness tests. You can review the updated methodology and results in sections "3. Materials and Methods" and "4. Results," which now span from pages 12 to 29.

  1. In its present form, your section 4.3 is oversimplified. For example, the effect of pension fund, sovereign wealth fund, and insurance company is always positive …

R: Thank you for pointing this out. As previously mentioned, we changed the model and the corresponding results. Thus, we had to update the results section. Some examples of the new text of results include:

Table 5 presents the results of the Random-effects ordered logistic regression. Wald tests of the set of variables indicate that the coefficients are significantly not jointly equal to zero (Fstat. = 736.78, p\0.000). This means that the model's fit is statistically significant. The signs of the estimated coefficients are interpreted as the ordered log odds of ECSI. A positive coefficient suggests that an increase in the explanatory variable is associated with a higher likelihood of moving to a higher category of ECSI. Conversely, a negative coefficient indicates a decrease in the likelihood of moving to a higher category.  (Page 21, line 1)

4.1. Financial variables result

The inclusion of Company Market Capitalization helped to control the size bias on the environmental rating presented [62, 63], showing that higher values of this variable would increase the probability of being in the superior categories. Results from the marginal effects show in Table 6 that the effect of being in each category is statistically significant at 99% of confidence in each category. The magnitude is higher according to its direction in the extreme categories. In category 5, Company Market Capitalization has a marginal probability of being in that category of 10% which decreases as we move down the categories until category 0, where the marginal probability of being is -5%.

The control variable "Total Debt Percentage to Total Equity," which represents leverage, exhibits a similar pattern of results but with a significantly lower magnitude. Marginal effect results indicate that the effect of each category is statistically significant with a confidence level of 99%. Additionally, for every increase of 4.09e-06 in "Total Debt Percentage to Total Equity," the marginal probability of being in the superior category increases, while the probability of being in the remaining categories decreases.

In the case of the profitability variable "ROA Margin," it exhibits a contrasting result compared to "Market Capitalization" and "Total Debt Percentage to Total Equity." "ROA Margin" is also statistically significant at a 99% confidence level across all categories. However, a 1% increase in "ROA Margin" will lead to a decrease of -0.1% in the probability of being in category 5. Conversely, a 1% increase in "ROA Margin" will increase the probability of being in the other categories. The lower categories show larger effects, with category 0 having the highest magnitude, resulting in a probability increase of 0.09% with a 1% rise in "ROA Margin."

4.2. Industry variables results

The focus of these empirical study analysis results is on determining whether the explanatory factors have positive or negative impacts. In this regard, the findings demonstrate that the use of control variables lowers the likelihood of analysis-related bias, such as bias based on the sector to which the company belongs. First off, the majority of industry variables have a considerable impact on each category and share similar behavior. 

This behavior implies that an increment would increase the probability of companies belonging to the lower categories while decreasing the probability of being in the superior categories. For instance, the Construction industry exhibits a 4% probability in category 0 and a -9% probability in category 5; the Information industry shows a 10% probability in category 0 and a -21% probability in category 5; the Professional, Scientific, and Technical Services industry demonstrates a 9% probability in category 0 and a -19% probability in category 5; the Real Estate, Rental, and Leasing industry has a 9% probability in category 0 and a -18% probability in category 5; the Quarrying, and Oil and Gas Extraction industry exhibits an 8% probability in category 0 and a -17% probability in category 5; the Retail Trade industry shows a 5% probability in category 0 and a -12% probability in category 5; the Transportation and Warehousing industry demonstrates an 8% probability in category 0 and a -18% probability in category 5; the Utilities industry has a 3% probability in category 0 and a -6% probability in category 5; the Finance & Insurance industry shows a 15% probability in category 0 and a -3% probability in category 5; and finally, the Wholesale industry exhibits a 7% probability in category 0 and a -15% probability in category 5. These results indicate that the industry to which a company belongs complements the variable of Company Market Capitalization, thereby correcting the effect of size bias.

4.3. Ownership variables result

The results indicate that the relationship between stakeholder ownership and ECSI can be categorized into three groups. The first group comprises Hedge Funds, Individual Investors, and Investment Advisors. Higher percentages of stocks owned by these investor types are associated with an increased probability of falling into the inferior categories of ECSI, while the probability of being in the superior categories decreases. Hedge Funds show statistical significance in each category, with positive and substantial probabilities of 0.2% and 0.1% in categories 0 and 2, respectively, and a negative impact of -0.4% in category 5. Similarly, Individual Investors exhibit a similar pattern, being statistically significant in each category, with a notable probability of 0.01% in category 0 and a negative marginal probability of -0.2% in category 5. In the case of Investment Advisors, their influence is of lower magnitude compared to the previous investor types. They demonstrate a marginal probability of 0.05% for category 0 and -0.1% for category 5.

On the contrary, Sovereign Wealth Funds exhibit a positive association with environmental supply chain performance, wherein a higher percentage of stocks owned by this investor type increases the probability of belonging to superior categories of ECSI and reduces the probability of falling into inferior categories. Specifically, in category 5, Sovereign Wealth Funds have a marginal probability increase of 0.1% for each share percentage held, while the remaining categories experience a decrease in marginal probability.

Finalmente, el tercer grupo no tiene relación estadística con ninguna categoría en la que se puedan encontrar tipos de inversionistas tales como Corporación, Compañía_de_Seguros y Fondo_de_Pensiones. Se pudo encontrar en la literatura que este tipo de inversionistas tienen una fuerte relación en su comportamiento de inversión con las regulaciones regionales. Por lo tanto, existe una oportunidad para que la investigación futura se concentre en cómo esta regulación afecta el interés de inversión de este inversor en ESG y prácticas ambientales de la cadena de suministro. (Página 24 a 25)

Reviewer 3 Report

The authors should provide a clearer explanation of the use and benefits of employing the statistical model. Additionally, it would be beneficial to include graphs alongside the existing table to enhance the clarity and provide further support for the research.

The references provided are sufficient to support the research. However, it would be helpful if the authors could elaborate on the sources or make the database used for the analysis publicly available.

Furthermore, the calculation methods for the variables and their corresponding sources are currently undisclosed.

Author Response

Reviewer 3

 The authors should provide a clearer explanation of the use and benefits of employing the statistical model. Additionally, it would be beneficial to include graphs alongside the existing table to enhance the clarity and provide further support for the research. The references provided are sufficient to support the research.

 R: Thank you for your comment. Indeed, we made improvements to the model and the explanation of the results in this new version of the paper. We included more graphics to give more strength to the explanations and conclusions.

 However, it would be helpful if the authors could elaborate on the sources or make the database used for the analysis publicly available. Furthermore, the calculation methods for the variables and their corresponding sources are currently undisclosed.

R: Thanks for your comment. Due to copyright issues, it is not possible to make the database used for the analysis publicly available. However, we are sharing with the Editor and Reviewers the links to the complete dataset used in the paper to allow further revision and we will share those links with any other researcher upon request. We had also clarified the calculation methods used in the paper to facilitate replicability of the research.

Reviewer 4 Report

No comments.

Author Response

Thanks for taking the time to review our paper.

Round 2

Reviewer 2 Report

See scores above.

Reviewer 3 Report

Thank you for your adjustment in your manuscript. The Manuscript is contribution to development enterprises.